# ConDS: Context Distribution Shift for Robust In-Context Learning

## Abstract

In-context Learning (ICL) is a popular approach to filling Large Language Models (LLMs) with the context without fine-tuning. ICL works by feeding the test input along with the context information selected from the candidate dataset as examples of explaining the target task and getting the answer. In real-world applications, noisy samples are easily to be included in the datasets, so it is unavoidable that the candidate set might also contain noise caused by human or measurement errors. The effectiveness of ICL is highly dependent on the quality of the selected ICL samples: the noise in the candidate set can mislead the query answer and severely degrade the ICL performance. However, the noise ICL problem is largely overlooked. To tackle this challenge, in this paper, we propose Context Distribution Shift (**ConDS**), which iteratively revises the distribution of the candidate dataset so that the retrieved ICL samples are emphasized to improve the robustness of ICL. Specifically, we first identify the clean and informative samples based on the retriever ranking score and the feedback from the LLMs, and then augment the identified informative samples. A subsampling strategy is adopted to emphasize the importance of informative samples and reduce the ratio of noisy samples. Thus, ICL's reliability can be improved by reducing the catastrophic impact of noisy samples on almost all test queries to a small percentage. Our **ConDS** can be easily combined with existing off-the-shelf and fine-tuned retrievers. An analysis is also provided to reveal the relationship between **ConDS** and retrievers. Experimental results show that **ConDS** outperforms baselines on various tasks under the influence of noise by a large margin of $8.12\%$.

## 1 Introduction

Large language models (LLMs) (Achiam et al., 2023; Touvron et al., 2023a;b; Team et al., 2023) have exhibited remarkable capabilities across a range of natural language processing and reasoning tasks (Wang et al., 2018; 2019). However, directly applying these LLMs to specific tasks can be challenging without task-specific adaptations due to the computational challenges of fine-tuning their vast number of trainable parameters. In-Context Learning (ICL) (Dong et al., 2022) represents a prominently efficient and effective way to utilize LLMs. Essentially, ICL operates by presenting LLMs with a set of selected ICL examples relevant to the test query from the candidate dataset $C$, preconditioning the models for the target task.

However, a real-world dataset, including the candidate set collected for ICL, can easily contain noisy samples. Given the critical dependence of ICL on the label quality of selected samples (Kossen et al., 2024; Wei et al., 2023), noise within the candidate set can significantly distort responses for the query from LLMs. Surprisingly, the issue of noise in ICL remains largely overlooked. There are two categories of existing ICL methods targeted at clean ICL settings. One adopts off-the-shelf retrievers (such as sparse retriever BM25 (Robertson et al., 2009) and dense retriever (Rubin et al., 2021)) to calculate the similar score between the given query and the candidate samples, and then retrieve ICL samples with the highest similarity scores. Another kind is fine-tuned retrievers trained on the specific tasks using the candidate datasets such as PromptPG (Lu et al., 2022), EPR (Rubin et al., 2021), etc. However, without additional treatments, the noisy samples are easily included in the ICL sample set retrieved by these methods, which finally misleads the query answer. Therefore, developing strategies to mitigate noisy information in the ICL candidate set becomes imperative.

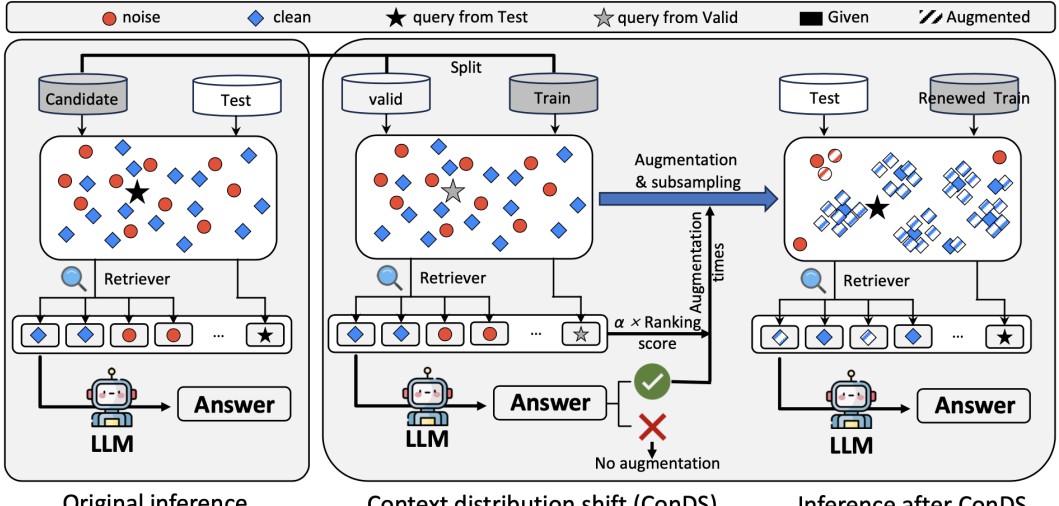

Figure 1: An overview of **ConDS**: Initially, we split the noisy candidate set into a training and validation dataset. Then, we evaluate the informativeness of the samples based on the LLM feedback of the validation samples and the retriever ranking score. Utilizing these scores, we reconstruct the training dataset and then resample it to emphasize the most informative samples and reduce the influence of the less useful samples. The revised training set is used for inference.

To tackle this challenge, instead of developing a new retriever, we focus on directly improving the quality of the candidate dataset by shifting its original distribution. Generally, clean samples are often informative for certain queries, while noisy samples can be mostly misleading. Our intuition is to augment the candidate set to increase the probability of informative samples being selected and decrease the probability of misleading samples being selected for test queries. To achieve this goal, we have to solve the following questions: *How to identify informative samples? How to change the distribution of the candidate sets?*

To solve these two questions, in this paper, we propose **C**ontext **D**istribution **S**hift (**ConDS**), which seeks to revise the distribution of the candidate dataset to improve the robustness of ICL. The framework of **ConDS** is shown in Figure 1. We split the candidate set into a training set and a validation set. We then identify informative samples in the training set based on the retriever ranking score of the chosen ICL samples and the LLM's feedback on the validation samples. The samples with positive feedback from LM will be considered informative samples and will be augmented. The augmentation strength is decided by the retriever ranking score. A subsampling strategy is adopted after augmentation to reinforce the importance of informative samples and decrease the influence of noisy samples. We iterate over all validation samples and repeat this process for a few epochs to renew the distribution. According to the experimental results on various tasks under the noise influence, our **ConDS** significantly outperformed baselines under different settings by decreasing the percentage of test queries affected by noisy samples from all to a small portion.

Our contributions are summarized as follows:

- We propose **ConDS**, which improves the quality of the candidate set by not only emphasizing informative samples but also reducing the impact of noisy label samples. We are the first to investigate the power of distribution shift of the candidate set to improve the ICL performance.

- **ConDS** supports different kinds of off-the-shelf and fine-tuned retrievers to enhance their robustness against noisy samples. We also provide an analysis to reveal the essential commonality between **ConDS** and the existing retrievers.

- Extensive experimental results on various benchmarks show that **ConDS** is robust to the noisy candidate dataset and significantly outperformed the baselines.

**Problem formulation of noisy ICL** Suppose we have a candidate sample pool $C = \{(x_i, y_i)\}_{i=1}^{N}$, among which $p \in (0, 1)$ ratio of samples have noise labels. Given a query $x_{test}$, we select $K$ in-

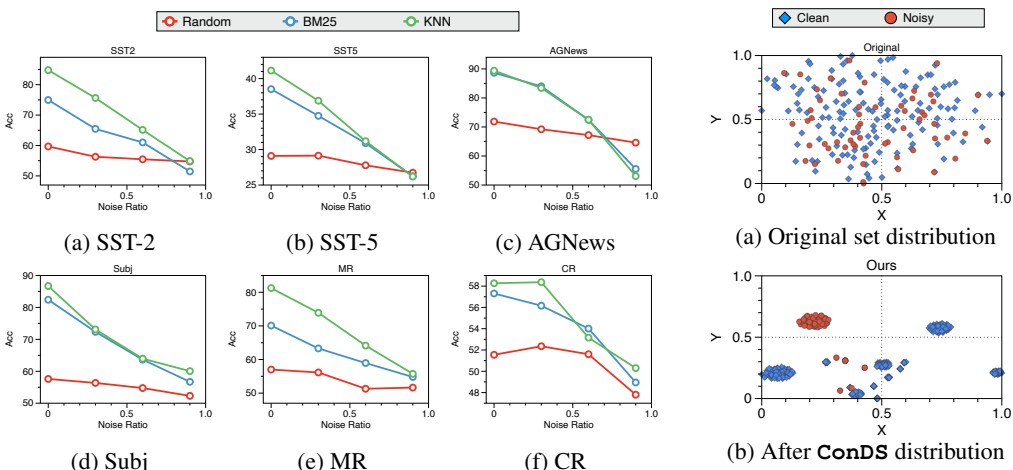

Figure 2: Effect of noise candidate sets on different datasets

Figure 3: Embedding visualization

context samples $E = \{(x_i, y_i)\}_{i=1}^{K}$ from the noise set $C$ and concatenate with the test query. The answer to the query is given by the LLM using

$$\hat{y}_{test} = \arg\max_{y \in \boldsymbol{y}} p_{\text{LLM}}(y|x_1 \oplus y_1 \cdots x_K \oplus y_K \oplus x_{test}), \tag{1}$$

where $\boldsymbol{y}$ is the label space and $\oplus$ denotes concatenation.

## 2 MOTIVATION: DISTRIBUTION SHIFT FOR CLEANING THE NEIGHBOR OF QUERIES

For real world applications, noisy samples are easily to be included in the datasets, so it is unavoidable that the candidate set collected for ICL might also contain noise. The effectiveness of the ICL is highly dependent on the quality of the candidate sets from which the in-context samples are drawn. The existence of the noisy samples in the candidate sets can significantly bring down the ICL performance even with a small noise ratio for various retrievers as shown in Figure 2 on 6 different datasets with 8-shots.

According to the t-SNE visualization of embedding in Figure 3a, the clean and noisy samples are mixed in the candidate set. During ICL, the retriever tends to select similar samples to the query as the ICL samples. With mixed clean and noisy samples, sampling similar samples using the retriever easily includes both clean and noisy samples for almost all query samples. The noisy samples with a false label can easily mislead the answer to the query, which leads to a significant performance drop, as shown in Figure 2.

Our intuition is to augment the candidate sets with more informative samples by adjusting the distribution of the candidate set. In most cases, clean samples are often informative for certain queries, while noisy samples can be mostly misleading and degrade the ICL performance. Our goal is to augment the neighbors of queries with more clean samples, increasing the retriever's probability of selecting clean samples instead of noisy ones. This intuition motivates the main challenges to be solved in this paper: *How to identify informative samples? How to change the distribution of the candidate sets?*

## 3 CONDS: CONTEXT DISTRIBUTION SHIFT

To sanitize the neighbor of the queries, in this section, we propose a context distribution shift method, **ConDS**, which identifies informative samples among the noise candidate set and shifts the distribution of the candidate set by augmenting the informative samples. The proposed **ConDS** can be combined with both the off-the-shelf retrievers such as KNN or BM25 (Robertson et al., 2009), and fine-tuned

retrievers such as PromptPG (Lu et al., 2022). We also elucidate the underlying relationship between **ConDS** and the retrievers through theoretical analysis.

The hypothesis of *query performance prediction* (QPP) (Bahri et al., 2020; Datta et al., 2022) states that similar queries should have similar retrieval effectiveness, which suggests that similar queries should have similar informative samples and misleading samples sampling from the candidate sets. Thus, we first randomly split the candidate set $C$ into a training set $C^{\text{train}} = \{(x_i^{\text{train}}, y_i^{\text{train}})\}_{i=1}^{N^{\text{train}}}$ and a validation set $C^{\text{valid}} = \{(x_i^{\text{valid}}, y_i^{\text{valid}})\}_{i=1}^{N^{\text{valid}}}$, and then use the feedback from the LLM on the validation set as the criterion for identifying informative (clean) and misleading (noise) samples. The shifted candidate sets can then be adopted for the unseen test samples based on the hypothesis.

## 3.1 ConDS for off-the-shelf retriever with static augmentation

We first introduce a naive version of **ConDS** for changing the distribution of candidate samples using the off-the-shelf retriever. Given a query $q_i = (x_i^{\text{valid}}, y_i^{\text{valid}}) \in C^{\text{valid}}$ from the validation dataset $C^{\text{valid}}$, the selected ICL set sampled from $C^{\text{train}}$ by the retriever $R$ is

$$E_i = \{e_i^k | e_i^k \sim R(e_i|q_i, C^{\text{train}})\}, \quad \text{for} \quad k = 1, 2, \cdots, K, \quad (2)$$

where $e_i^k = (x_i^k, y_i^k)$. Following Eq. (1), the selected ICL sample set $E_i$ is then concatenated with the query $x_i^{\text{valid}}$ and fed into the LLM, the answer of $q_i$ is generated by

$$\hat{y}_i^{\text{valid}} = \arg\max_{y \in \boldsymbol{y}} p_{\text{LLM}}(y|x_i^1 \oplus y_i^1 \cdots x_i^k \oplus y_i^k \cdots x_i^K \oplus y_i^K \oplus x_i^{\text{valid}}), \quad (x_i^k, y_i^k) \in E_i. \quad (3)$$

An evaluation is made between $\hat{y}_i^{\text{valid}}$ and the ground truth $y_i^{\text{valid}}$. The reward for each validation sample is given by

$$\text{EVAL}(\hat{y}_i^{\text{valid}}, y_i^{\text{valid}}) = \mathbb{1}(\hat{y}_i^{\text{valid}}, y_i^{\text{valid}}). \quad (4)$$

If $\text{EVAL}(\hat{y}_i^{\text{valid}}, y_i^{\text{valid}}) = 1$, we consider the samples selected for this query sample as informative samples. We augment the ICL samples in $E_i$ selected for this validation query $q_i$ adopting either directly duplicating or existing paraphrasing methods (Vladimir Vorobev, 2023)[1]. We define the duplicating/ paraphrasing times per sample in $E_i$ as $\alpha$. Then the original ICL set $E_i = \{e_i^k\}$ for $q_i$ transforms into

$$E_i^{\text{shift}} = \{e_i^{k,j}\}, \quad \text{for} \quad k = 1, 2, \cdots, K \quad \text{and} \quad j = 1, 2, \cdots \alpha, \quad (5)$$

where $e_i^{k,j}$ are directly duplication or paraphrase of $e_i^k$. Since $E_i \subseteq C^{\text{train}}$, the original $E_i$ in $C^{\text{train}}$ is then replaced by the augmented set $E_i^{\text{shift}}$. To avoid the training set $C^{\text{train}}$ growing too large and thus affecting the efficiency of the inference time, we subsample $C^{\text{train}}$ if its size reaches an upper limit $N_{\text{upp}}$. We iterate over all validation samples in $C^{\text{valid}}$ and repeat this augmentation process. The shifted training set is then adopted during the inference stage.

After several iterations of augmentation, the most informative samples will be selected and augmented most frequently with a positive evaluation reward, while the misleading samples are selected with lower frequency, even if they are selected, the zero reward will not lead to an augmentation of these kind of samples. The original distribution of $C^{\text{train}}$ shifts from the original one, and the most informative sample size grows much larger while the misleading sample size grows smaller or even disappears due to the subsampling strategy.

As shown in Figure 3a, using the original $C^{\text{train}}$, since the noise and clean samples are mixed, during the inference stage, the retriever will select both noise and clean samples for almost all test queries. After **ConDS**, as shown in Figure 3b, instead of mixing clean and noisy samples, the neighbors of the clean samples are also augmented with more clean samples. During the inference stage, the retriever tends to select the most relevant samples for the test queries. The most relevant spaces are filled with clean samples, and the misleading samples tend to have a lower relevance score. Misleading sample embeddings stay far away from the clean samples cluster, so they will not interfere with the test queries lying close to the clean samples. Hence, we reduce the catastrophic impact of the noisy samples from almost all test queries to only a small percentage of queries[2]. We provide experimental results to verify this point in Section 4.3.

---

[1]We note that the augmentation method itself is not the focus of this paper, where many existing methods can be plug in. This paper focuses on *what samples* should be augmented.

[2]noisy samples still exist in Figure 3b due to noise in the validation dataset.

## 3.2 ConDS for fine-tuned retriever with dynamical augmentation

**ConDS** can also be combined with the training stage of the reinforcement-based fine-tuned retriever such as PromptPG (Lu et al., 2022). In this subsection, we will focus on discussing the difference compared with Section 3.1. Given the training set $C^{\text{train}}$ and the query $q_i$, the fine-tuned retriever is $R(e_i|q_i, C^{\text{train}}, w)$, where $w$ is the parameter for $R$. The selected ICL sample set for $q_i$ is

$$E_i = \{e_i^k | e_i^k \sim R(e_i|q_i, C^{\text{train}}, w)\}, \quad \text{for} \quad k = 1, 2, \cdots, K. \tag{6}$$

The ranking score returned by the retriever $R$ for all the ICL samples is

$$S_i = \{s_i^k | s_i^k \sim \text{SCORE}(e_i^k|q_i, C^{\text{train}}, w)\}, \quad \text{for} \quad k = 1, 2, \cdots, K. \tag{7}$$

For each trained epoch, once the reward for the evaluation (Eq. (4)) is positive, the retriever will be updated to $R(e_i|q_i, C^{\text{train}}, w')$ using its own training strategy, as a result, the ICL sample set $E_i$ and the returned ranking score $S_i$ will also be updated correspondingly. Given a pre-defined augmentation hyperparameter $\alpha$, the augmentation size for the selected ICL samples $E_i$ will be $\alpha S_i = \{\alpha s_i^k\}$. Then the original $E_i = \{e_i^k\}$ transforms into

$$E_i^{\text{shift}} = \{e_i^{k,j}\}, \quad \text{for} \quad k = 1, 2, \cdots, K \quad \text{and} \quad j = 1, 2, \cdots \alpha s_i^k, s_i^k \in S_i. \tag{8}$$

Then the original $E_i$ in $C^{\text{train}}$ is replaced by the renewed set $E_i^{\text{shift}}$. A subsampling strategy same as Section 3.1 is also adopted afterwards.

The augmentation size for different samples is dynamic due to different ranking scores for these samples, and the augmentation size for each training epoch will also change dynamically w.r.t. the updating of the retriever itself. After the training stage, the shifted set is used for the test inference. Note that **ConDS** will not introduce any additional token consumption for query compared with training a retriever w/o **ConDS**. The algorithm is summarized in Algorithm 1.

---

**Algorithm 1 ConDS** for fine-tuned retriever

---

1: **Input**: Retriever $R$, language model LLM, candidate set $C$, upper limit $N_{\text{upp}}$ for $|C|$, epochs $T$;
2: **Output**: The context shift candidate set $C^{\text{train}}$.
3: Randomly split candidate set $C$ into $C^{\text{train}}$ and $C^{\text{valid}}$.
4: **for** $t = 1, \ldots, T$ **do**
5:    **for** each query sample $q_i = (x_i^{\text{valid}}, y_i^{\text{valid}})$ in $C^{\text{valid}}$ **do**
6:       Retrieve the selected ICL sample set $E_i$ for $q_i$ using Eq. (6).
7:       Concatenate query question $x_i^{\text{valid}}$ and $E_i$, get the answer $\hat{y}_i^{\text{valid}}$ from LLM using Eq. (3).
8:       Calculate the evaluation reward $\text{EVAL}(\hat{y}_i^{\text{valid}}, y_i^{\text{valid}})$ using Eq. (4).
9:       **if** $\text{EVAL}(\hat{y}_i^{\text{valid}}, y_i^{\text{valid}}) = 1$ **then**
10:          Upadate retriever $R$ using its own training strategy.
11:          $E_i \leftarrow E_i^{\text{shift}}$   ▷ using Eq. (8) based on the updated ranking score (Eq. (7)).
12:          $C_{t-1}^{\text{train}} \leftarrow C_t^{\text{train}}$   ▷ replacing the original $E_i$ in $C^{\text{train}}$ with $E_i^{\text{shift}}$.
13:       **end if**
14:    **end for**
15:    **if** $|C_t^{\text{train}}| > N_{\text{upp}}$ **then**
16:       $C_t^{\text{train}} \leftarrow \text{Random\_sample}(C_t^{\text{train}}, N_{\text{upp}})$.
17:    **end if**
18: **end for**

---

## 3.3 Analysis for the relationship between ConDS and the retriever

We provide an analysis to reveal the relationship between the candidate set distribution shift and the retriever in this section.

**Lemma 1.** *The distribution shift of candidate samples can be transformed into a fine-tuned retriever with the sampling probability as the ranking score.*

For each epoch $t$, we augment all the samples in training set $C_{t-1}^{\text{train}}$ from last epoch based on Eq. (8), so the number of augmentation size for $e^k \in C_{t-1}^{\text{train}}$ will be the summation of the scores from

all the validation data: $M = \sum_{i=1}^{N} \alpha s_i^k \mathbb{1}(\hat{y}_i^{\text{valid}}, y_i^{\text{valid}}), s_i^k \in S_i$, where $N$ is the number of the validation data. The number of the entire candidate set for epoch $t$ will be $N_t^{\text{train}} = |C_t^{\text{train}}| = \sum_{k=1}^{N_{t-1}^{\text{train}}} \sum_{i=1}^{N} \alpha s_i^k \mathbb{1}(\hat{y}_i^{\text{valid}}, y_i^{\text{valid}})$. According to the theory of hypergeometric distribution (AA, 1995), after random sampling $N_{\text{upp}}$ samples from $C_t^{\text{train}}$, the probability of $e^k$ to be sampled for epoch $t$ is

$$P_t(e^k) = 1 - Pr(|\{e^{k,j}\}| = 0 | e^{k,j} \in C_t^{\text{train}})$$

$$= 1 - \frac{\binom{M}{0}\binom{N_t^{\text{train}} - M}{N_{\text{upp}} - 0}}{\binom{N_t^{\text{train}}}{N_{\text{upp}}}} = 1 - \frac{\binom{N_t^{\text{train}} - M}{N_{\text{upp}}}}{\binom{N_t^{\text{train}}}{N_{\text{upp}}}} \equiv 1 - \xi(e^k).$$

The noisy samples tend to have a large probability of having a 0 reward ($\mathbb{1}(\hat{y}_i^{\text{valid}}, y_i^{\text{valid}})$), and the score $s$ returned from the retriever is also low due to its weaker correlation with the validation queries. As a result, the $M_{\text{noisy}}$ for noisy samples tends to be much smaller than $M_{\text{clean}}$ for clean samples. Thus, for noisy samples, $M_{\text{noisy}} \ll N_t^{\text{train}}$, $\xi(e^k) \to 1$, and $P_t(e^k) \to 0$. $P_t(e^k)$ for noisy samples will even grow smaller with the increase of epochs $t$. On the contrary, clean samples have a higher probability $P_t(e^k)$ of being kept with a larger $M$. In this way, the probability $P_t(\cdot)$ can serve as the ranking score by giving a higher score for clean samples and a lower score for noisy samples. With the increase of epoch $t$, $P_t(\cdot)$ dynamically changes for each candidate sample, which can be considered as dynamically fine-tuning a retriever.

Due to the essential commonality between **ConDS** and the retriever, **ConDS** can be flexibly combined with the existing retrievers to amplify their effectiveness in selecting clean samples. By combining the two ranking scores, the hybrid score for candidate sample $e^k$ becomes $P_t(e^k)s_i^k$ where $i$ indicates the $i$-th test query.

## 4 EXPERIMENT

In this section, we first introduce the experiment setting and then verify the effectiveness of **ConDS**.

### 4.1 EXPERIMENT SETTING

**Datasets.** We conduct experiments on a wide range of tasks: 1) Sentiment Classification: SST-2, SST-5 (Socher et al., 2013), MR (Pang & Lee, 2005), CR (Kim Amplayo et al., 2022); 2) Topic Classification: AGNews (Zhang et al., 2015), TREC (Voorhees & Tice, 2000); 3) natural language inference: MNLI (Williams et al., 2017), RTE (Bar-Haim et al., 2014); 4) Subjectivity Classification: Subj (Pang & Lee, 2004).

**Baselines.** We compare our method with the following baselines: *1) Zero-shot:* Only the test query is fed into the LLM, 0 ICL sample is selected. *2) Random:* We randomly sample ICL samples from the candidate set. *3) BM25* (Robertson et al., 2009) is an off-the-shelf sparse retriever. Given a test query, BM25 can retrieve the most relevant samples from the candidate set with a similar input as the test query. *4) KNN* (Reimers & Gurevych, 2019) utilizes the Sentence-BERT as the off-the-shelf dense demonstration retriever. It uses "paraphrase-mpnet-basev2" (Rubin et al., 2021) to encode the test query and candidate set's inputs. The examples with the most similar input are selected as the ICL samples. *5) DPP* (Chen et al., 2018) uses BERT-based embedding and adopts MAP inference for retrieving relevant samples from the candidate set. *6) PromptPG* (Lu et al., 2022) is a fine-tuned retriever, which utilizes policy gradient to fine-tune a BERT-based (Devlin et al., 2018) retriever to learn to select ICL samples for the test query.

**Noise setting.** We inject noise in the ICL Database by coin tossing with probability $p$. Any sample in the original dataset has a probability of $p$ being changed to another false label. $p$ is set to 0.6 by default.

**Implementation.** We adopt GPT-Neo-2.7B (Black et al., 2021) as the inference LLM by default. The shot number is $K = 20$, and the candidate set size is 200 by default. For fine-tuned retrievers, we randomly split $10\%$ of the candidate data as $C^{\text{valid}}$. Training epochs are set as 5, the learning rate is $1e - 4$, the augmentation parameter $\alpha$ is 1000, and the augmentation method is set as direct duplication unless otherwise mentioned.

| Retrieval Method | Dataset | | | | | | | | | Avg |
|---|---|---|---|---|---|---|---|---|---|---|
| | SST-2 | SST-5 | AGNews | Subj | MR | CR | TREC | RTE | MNLI | |
| Zero-shot | 0.7359 | 0.2919 | 0.6760 | 0.5055 | 0.7395 | 0.6207 | 0.3140 | 0.4874 | 0.3550 | 0.5251 |
| Random | 0.5656 | 0.2602 | 0.7240 | 0.6460 | 0.5285 | 0.5326 | 0.4560 | 0.5632 | 0.3327 | 0.5121 |
| BM25 | 0.5338 | 0.2765 | 0.7363 | 0.6935 | 0.5485 | 0.5511 | 0.5180 | 0.4874 | 0.3437 | 0.5210 |
| KNN | 0.5634 | 0.2945 | 0.7380 | 0.6990 | 0.5600 | 0.5531 | 0.5140 | 0.5415 | 0.3437 | 0.5341 |
| DPP | 0.5327 | 0.2407 | 0.4663 | 0.6390 | 0.5160 | 0.5175 | 0.3580 | 0.5379 | 0.3177 | 0.4584 |
| PromptPG | 0.6870 | 0.4235 | 0.7687 | 0.7340 | 0.8005 | 0.6553 | 0.4880 | **0.5740** | 0.4007 | 0.6146 |
| PromptPG+**ConDS** (duplicate) | **0.8479** | 0.4579 | 0.7530 | **0.7905** | **0.9045** | **0.9108** | 0.5380 | 0.5560 | **0.5040** | **0.6958** |
| PromptPG+**ConDS** (paraphrase) | 0.7760 | **0.4887** | **0.8107** | 0.7735 | 0.8200 | 0.8877 | **0.5740** | 0.5487 | 0.4147 | 0.6771 |

Table 1: Evaluation results on various baselines and **ConDS**. The average performance of three random seeds for each experiment is reported. The best performance for each dataset is highlighted in **bold** font and the second-best performance is underlined.

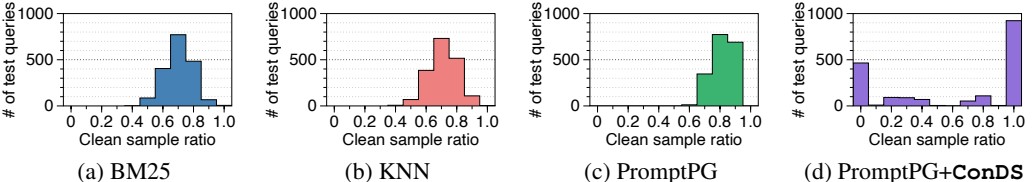

| (a) BM25 | (b) KNN | (c) PromptPG | (d) PromptPG+**ConDS** |
|---|---|---|---|

Figure 4: The distribution of the clean sample ratio of selected ICL samples of test queries for different retrievers and **ConDS**. For **ConDS**, $50.25\%$ of the test queries have $100\%$ clean samples in the selected ICL set, while for other baselines, $0\%$ test queries have $100\%$ clean samples.

## 4.2 MAIN RESULTS

We compare our proposed **ConDS** with the baselines on various tasks under the influence of noise in Table 1. The results show that with noisy samples included in the candidate set, the 20-shot results for baselines even perform worse than zero-shot learning on 5 datasets. One possible reason is that many noisy samples are selected as ICL samples and mislead the final prediction results. **ConDS** (duplicate) outperformed zero-shot learning by $17.07\%$, and the best baseline by $8.12\%$ on average. **ConDS** (parahprase) outperformed zero-shot learning by $15.20\%$, and the best baseline by $6.25\%$ on average. These results demonstrate that **ConDS** can significantly reduce the catastrophic impact of the noisy samples and improve the robustness of ICL.

**Effects of different augmentation methods.** We also investigate the effects of two augmentation methods of **ConDS** including direct duplication and paraphrase. Paraphrase (Vladimir Vorobev, 2023) adopts a T5-based model trained on a ChatGPT paraphrase dataset. According to the results, paraphrase achieves better ICL performance than duplication on three datasets, and duplicate achieve better ICL performance on five datasets. For most datasets, both augmentation method achieves either the best or second-best performance. These results indicate that the distribution shift induced by **ConDS** can improve the robustness of ICL no matter what augmentation method is adopted. Since the main focus of this paper is the distribution shift of the candidate set instead of the augmentation method itself, we leave the exploration for other augmentation methods for future works. We use duplication as the default augmentation method for the following experiments.

## 4.3 QUALITATIVE STUDIES

To further verify our hypothesis in Section 3.1 that the improvement of **ConDS** is caused by cleaning the neighbor of some of the queries so that we can reduce the catastrophic impact of the noisy samples from almost all test queries to only a small percentage of queries, we investigate **the clean sample ratio of selected ICL samples for test queries** w.r.t different methods in Figure 4 on SST-2. According to the results, for BM25, KNN, and PromptPG, $0\%$ of the test queries have $100\%$ clean samples, which indicates that noisy samples exist in all selected ICL sets for all the test queries. The reason for this phenomenon is that for the original candidate set, the clean and noisy samples are mixed with each other, as shown in the embedding distribution visualization in Figure 3a. When we adopt ICL to select the most relevant samples to the queries, both noise and clean samples can be retrieved.

For retriever w/ **ConDS**, $50.25\%$ of the test queries have $100\%$ clean samples in the selected ICL set, since we augment informative samples and clean the neighbor of test queries as shown in Figure 3b. Compared with $0\%$ of other baselines, we significantly reduce the noisy samples' catastrophic impact on the queries. The percentage for lower clean sample ratios also increases due to the noisy samples in the validation dataset, but this will not have a significant impact on the performance, as we observe that even a small percentage of noisy samples in the selected ICL sample set has a chance to mislead the query answer, as long as the percentage is smaller than $100\%$. With clean sample ratio $\geq 0.7$, 512, 444, and 331 test queries were answered incorrectly for BM25, KNN, and PromptPG. Similar results can also be found in Figure 2, even with a small noise ratio $p$, accuracy drops significantly. To solve this problem, a higher percentage of queries with $100\%$ clean samples is more crucial. **ConDS** increases the robustness of ICL by significantly bringing the noisy sample impact on test queries to a lower percentage. To better verify the effectiveness of **ConDS**, we also provide case studies on the retrieved samples for different retrievers in Appendix D.

## 4.4 ABLATION STUDIES

**Effects of ConDS for retrievers.** We show the effects of different retrievers w/ **ConDS** in Table 2. For different retrievers, we can observe an average improvement of $1.26\%$, $3.36\%$, $5.54\%$, $6.83\%$, and $9.77\%$, respectively, which shows that our **ConDS** can be flexibly combined with different kinds of retrievers. The more capable the retriever is, the more boosts we get for the ICL performance. As analyzed in Section 3.3, our **ConDS** can be considered as a special kind retriever, and the hybrid ranking score for the combined retriever is $P_t(e^k)s_i^k$, where $P_t(e^k)$ is the sampling probability and $s_i^k$ is the scored returned by the original retriever. The hybrid ranking score amplifies the effect of the original retriever on selecting clean samples.

| Dataset | Retriever | w/o ConDS | w/ ConDS |
|---------|-----------|-----------|----------|
| SST-2 | Random | 0.5656 | **0.5677** |
| | BM25 | 0.5338 | **0.5721** |
| | KNN | 0.5634 | **0.6397** |
| | DPP | 0.5327 | **0.6485** |
| | PromptPG | 0.6870 | **0.8479** |
| SST-5 | Random | 0.2602 | **0.2833** |
| | BM25 | 0.2765 | **0.3054** |
| | KNN | 0.2945 | **0.3290** |
| | DPP | 0.2407 | **0.2615** |
| | PromptPG | 0.4235 | **0.4579** |

Table 2: Effects of retrievers w/ ConDS.

**Effects of different noise ratios.** The effect of different noise ratios on SST-2 and Subj is shown in Figure 5a. **ConDS** consistently outperforms all the baselines under the influence of different noise ratios ranging from 0.1 to 0.6. The average improvement for **ConDS** is $5.76\%$, $6.80\%$, and $6.85\%$ compared with the best baseline. Different noise ratios have the largest impact on off-the-shelf retrievers. With different noise ratios, BM25 and KNN have very unstable ICL performance, with an average accuracy drop of $12.14\%$ and $18.34\%$. The fine-tuned retriever PromptPG shows better stability with an average accuracy difference of $12.06\%$. Our proposed **ConDS** shows the best stability with an average accuracy drop of $9.12\%$, which verifies that different noise ratios have the smallest impact on **ConDS**.

**Effects of different candidate sizes.** We investigate the effects of different candidate sizes in Figure 5b on SST-2 and SST-5. **ConDS** consistently outperformed all the baselines w.r.t. different candidate sizes. The off-the-shelf retrievers are more stable, while the fine-tuned retriever is more sensitive. The fine-tuned retriever PromptPG first increases and then decreases with an average accuracy difference of $12.3\%$ between the best and worst results. Since PromptPG+**ConDS** is based on PromptPG, it has a similar accuracy trend as PromptPG, but PromptPG+**ConDS** is much more stable with an accuracy difference of only $6.14\%$. Combining the existing retrievers with **ConDS** decreases the impact of the candidate dataset size. Shifting the distribution of the candidate set and training the retriever simultaneously allows the retriever to adapt to different candidate data sizes.

**Effects of # of shots.** We investigate the effects of different shot # in Figure 5c. **ConDS** consistently outperformed other baselines for different shot numbers with an average improvement of $11.91\%$, $2.11\%$, and $9.77\%$, compared with the best baseline. The off-the-shelf retriever is not sensitive to the change of shot numbers but has lower accuracy. PromptPG has higher accuracy than the off-the-shelf retrievers but with more variations. Our proposed **ConDS** is the most stable one and achieves the best ICL performance.

**Effects of different augmentation parameter $\alpha$.** The effects of different $\alpha$ on SST-2 and SST-5 are shown in Figure 5d. The most suitable value of $\alpha$ is different given different datasets. If the value of $\alpha$ is too small, the augmentation for informative samples is not sufficient, and noisy samples

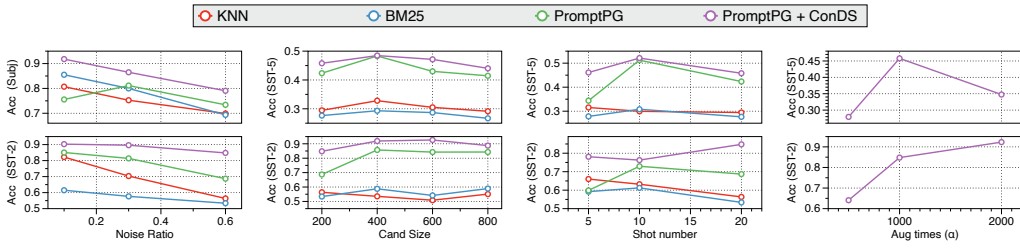

(a) Various noise ratios   (b) Candidate size analysis   (c) Various # of shots   (d) Aug parameter

Figure 5: Ablation studies on different noise ratios, candidate sizes, # of shots, and augmentation parameters $\alpha$.

| Dataset | SST-2 | | | | SST-5 | | | |
|---|---|---|---|---|---|---|---|---|
| LMs/retrievers | BM25 | KNN | PromptPG | PromptPG+ConDS | BM25 | KNN | PromptPG | PromptPG+ConDS |
| GPT2-XL | 0.5222 | 0.4355 | 0.8715 | **0.9006** | 0.2937 | 0.2910 | 0.3462 | **0.3670** |
| GPT-Neo-1.3B | 0.5321 | 0.6469 | 0.7683 | **0.9072** | 0.2882 | 0.3461 | 0.2457 | **0.4221** |
| GPT-Neo-2.7B | 0.5338 | 0.5634 | 0.6870 | **0.8479** | 0.2765 | 0.2945 | 0.4235 | **0.4579** |

Table 3: Transferability of **ConDS** and other baselines across different LMs.

can also exist in the augmented samples. The imbalance of noise and informative samples degrades the performance. If the value of $\alpha$ is too large, too much augmentation of some particular samples can decrease the diversity of the selected samples, so the accuracy might drop. We set $\alpha = 1000$ by default to avoid insufficient or excessive augmentation.

**Transferability of ConDS across different LMs.** We evaluate the transferability of **ConDS** across different language models including GPT2-XL (Radford et al., 2019), GPT-Neo-1.3B (Black et al., 2021), and GPT-Neo-2.7B (Black et al., 2021) on two datasets in Table 3. The results show that **ConDS** shows the best ICL performance for all the LLMs, which indicates that the robustness of **ConDS** is transferable across different sizes of LLMs. Even if the fine-tuned retriever PromptPG performs worse than the off-the-shelf retriever, as for GPT-Neo-1.3B on SST-5, PromptPG achieves only 24.57% accuracy, but combined with our proposed **ConDS** improves the accuracy to 42.21%.

**Training time cost.** The extra time during training caused by our **ConDS** is negligible. For SST-2, the training time for 1 epoch of PromptPG and PromptPG+**ConDS** is 12 and 13 seconds, respectively.

### 4.5 ConDS FOR GENERATION TASKS UNDER NOISY DATASET

We conduct the following experiments to not only evaluate the performance on classification tasks but also to further validate the effectiveness of LLMs in generation tasks.

| Question | Clean answer | Noise answer |
|---|---|---|
| What type of money does jamaica use? | Jamaican dollar | lady |
| When did the charlotte bobcats first play in the NBA? | 2004 NBA Draft | Reece Shearsmith |

Table 4: Examples of noisy samples in generation tasks. We randomly select answers from a corpus.

**Experiment setting.** We conduct experiments on Open-Domain Question Answering task WebQ (Berant et al., 2013) and Squad (Rajpurkar, 2016). We follow Gao et al. (2024) to inject noise by replacing the original answer with a random answer from a large corpus with probability $p$. Any sample in the original dataset has a probability of $p$ being changed to another false answer. Detailed examples are shown in Table 4, and the construction of the large corpus is presented in Appendix C. As shown in Table 4, the answers are not related to the topics. For instance, even though the question is about basketball, the provided answer in the example is irrelevant. We adopt Llama2-7B (Touvron et al., 2023a) to examine the generation tasks. Other settings follow the default ones in Section 4.1. Exact match (EM) is used as the evaluation metric for the generation task.

**Generation task results.** As described in Table 5, we can summarize the following findings. First, irrelevant noisy information can cause performance degradation for baselines, which becomes more severe as the noise ratio increases (e.g., $0.1521 \rightarrow 0.0529$ in BM25 case for WebQ). Second, as shown in **bold** font, **ConDS** demonstrates improved performance compared to other retrieval methods. For the case when the noise ratio is 0.4, our method outperformed the best baseline by 8% and

| Dataset | Noise ratio | Retrieval Methods | | | | |
|---|---|---|---|---|---|---|
| | | Random | BM25 | KNN | PromptPG | PromptPG+ConDS |
| WebQ | 0.2 | 0.1124 | 0.1521 | 0.1217 | 0.1250 | **0.1600** |
| | 0.4 | 0.0384 | 0.0529 | 0.0437 | 0.0850 | **0.1650** |
| Squad | 0.2 | 0.3040 | 0.2930 | 0.3100 | 0.3340 | **0.3590** |
| | 0.4 | 0.2390 | 0.2230 | 0.2140 | 0.2680 | **0.3870** |

Table 5: Evaluation results on various baselines and ConDS for noise generation tasks. The best performance for each dataset is highlighted in **bold** font and the second-best performance is underlined.

11.9%, respectively for two datasets. This indicates that focusing on clean samples can mitigate the performance decline caused by noise ICL samples. Moreover, even with an increased noise ratio $0.2 \rightarrow 0.4$, ConDS shows a stable performance $0.1600 \rightarrow 0.1650$ and $0.3590 \rightarrow 0.3870$ without a degradation. Consequently, utilizing ConDS is a robust method for both classification and generation tasks when ICL dataset has noise information.

## 5 RELATED WORK

**In-context learning.** One mainstream of ICL (Dong et al., 2022) utilizes off-the-shelf retrievers, which are classified into two types: sparse and dense retrievers. BM25 (Agrawal et al., 2022) is a sparse retriever that uses term-frequency scores to measure the relationship between a query and in-context example. The main weakness of this sparse retriever is understanding semantic information. To address this issue, dense retrievers have been adopted by employing neural networks to comprehend the meaning of sentences rather than individual words. Liu et al. (2021) utilize a BERT model to build a KNN-based retriever. Another main category is to fine-tune a prompt retriever to select examples on specific tasks. Rubin et al. (2022) train an efficient retriever that uses positive and negative information from the dense passage retriever (Karpukhin et al., 2020). UDR (Li et al., 2023) proposes a universal retriever applicable across various domain tasks. PromptPG (Lu et al., 2022) employs a reinforcement learning framework to train the retriever to find the most informative examples for answering. DATAMODELS (Chang & Jia, 2022) trains linear regressors according to the example influence on the LLM prediction. LLM-R (Wang et al., 2023) trains retrievers using a proposed reward model. Some works tried to retrieve ICL samples from other perspectives. Li & Qiu (2023) chooses the most representative examples for all test cases by using contribution measures. Xie et al. (2021) addresses the in-context learning problem by selecting appropriate sample problems as an implicit Bayesian inference.

**Noisy dataset with ICL.** Pan (2023) explore the embedded pre-trained knowledge in LLMs by substituting the label word with an arbitrary word, but they did not survey noisy (incorrect) labels. Kossen et al. (2024); Wei et al. (2023) examine the impact of noisy labels from the ICL perspective, considering various factors such as the number of ICL samples and the model size of LLMs. However, their investigation focuses on observing the phenomenon rather than proposing a solution to address the issue. Cheng et al. (2024) discover that introducing label noises during the training of the LM can improve the robustness of Transformers during ICL inference. However, training LLMs from scratch takes a lot of time and computational resources, and is inapplicable for black-box LMs (eg, GPT-3 (Brown et al., 2020), GPT-4 (Achiam et al., 2023)). Hence, in this paper, we focus on a black-box LLM setting and do not have access to the training of the LLM.

## 6 CONCLUSION

In this paper, we propose ConDS to improve the robustness of in-context learning (ICL) against noisy samples which has been reported in several researches. The main philosophy of the proposed algorithm is a context set distribution shift method. Briefly, the algorithm works as follows. First, ConDS identifies the informative samples based on the feedback from LLM and the ranking scores from the retriever, and then augment these informative samples. A sub-sampling strategy is also used to increase the probability of sampling clean samples and decrease the probability of sampling noisy samples. ConDS can be flexibly combined with both off-the-shelf and fine-tuned retrievers. Experimental results for various tasks including classification and generative tasks under noise setting show that ConDS significantly outperformed baselines.

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

-Supplementary Material-

# ConDS: Context Distribution Shift for Robust In-Context Learning

## A    EXPERIMENTAL DETAILS

Given that it is well-known that the input instruction prompt can significantly affect performance, we clarify the prompts used for each dataset Table 6. We adhere to the prompt settings outlined in Li et al. (2023) and also utilize the dataset uploaded by the author of this paper; `https://huggingface.co/KaiLv`.

| Dataset | Prompt | Label | Label template | Example |
|---------|--------|-------|----------------|---------|
| AGNews | Topic of the text: | {World, Sports, Business, Technology } | Topic: **Label** | REDMOND, Wash. - Microsoft Corp. and cable television provider Comcast Corp. said Monday they would begin deploying set-top boxes powered by Microsoft software starting next week. \n Topic: Business \\NEW YORK (Reuters) - Venus Williams advanced to the second round of the U.S. Open on Tuesday but had to work hard for her 6-3, 7-6 victory against Hungary's Petra Mandula.\n Topic: Sports ...Oil demand is rising faster than predicted this year as OPEC pumps more low-quality oil in a failed bid to reduce record prices, according to International Energy Agency, an adviser to 26 industrialized nations. \n Topic: |
| RTE | Recognizing textual entailment between these 2 texts. | {True, False} | Answer: **Label** | o Weapons of Mass Destruction Found in Iraq Yet. Question: Weapons of Mass Destruction Found in Iraq. Ture of False? \n Answer: False \\A place of sorrow, after Pope John Paul II died, became a place of celebration, as Roman Catholic faithful gathered in downtown Chicago to mark the installation of new Pope Benedict XVI. Question: Pope Benedict XVI is the new leader of the Roman Catholic Church. Ture of False? \n Answer: True ...Steve Jobs was attacked by Sculley and other Apple executives for not delivering enough hot new products and resigned from the company a few weeks later. Question: Steve Jobs worked for Apple. Ture of False?\n Answer: |
| MNLI | Recognizing textual entailment between these 2 texts. | {Entailment, Inconclusive, Contradiction} | Answer: **Label** | uh-huh exactly not what color you are how old you are what if your male or female that would be wonderful i guess it's kind of an ideal world though huh Based on that information, is the claim The world would be better if race and gender did not matter. People would get along much better "Entailment", "Contradiction", or "Inconclusive"? \n Answer: Inconclusive \\uh-huh exactly not what color you are how old you are what if your male or female that would be wonderful i guess it's kind of an ideal world though huh Based on that information, is the claim The world would be better if race and gender did not matter. "Entailment", "Contradiction", or "Inconclusive"? \n Answer: Entailment ...It's that kind of world. Based on that information, is the claim The world is getting better. "Entailment", "Contradiction", or "Inconclusive"?\n Answer: |

Table 6: Prompt and instruction used for each dataset. We denote examples as blue color, and query as red color, respectively.

| Dataset | Prompt | Label | Label template | Example |
|---------|--------|-------|----------------|---------|
| TREC | Topic of the question: | {Description, Entity, Expression, Human, Location, Number} | Topic: *Label* | How did serfdom develop in and then leave Russia ? \n Topic: Description \\What films featured the character Popeye Doyle ?\n Topic: Description ...Who developed the vaccination against polio ?\n Topic: |
| CR | Sentiment of the sentence: | {great, terrible} | It was *Label* | it 's not as stylized as a sony or samsung . \n It was terrible \\the 6600 will provide similar service in more developed areas of the states and not as well in more remote areas .\n it was terrible ...apex does n 't answer the phone .\n It was |
| SST-2 | Sentiment of the sentence: | {great, terrible } | It was *Label* | a string, funny and finally transportin re-imagining of beauty and the beast and 1930s horror films \n It was great \\apparently ressembled from the cutting-room floor of any given daytime soap. \n It was terrible ...no movement, no yuks, no much of anythin. \n It was |
| MR | Sentiment of the sentence: | {great, terrible} | It was *Label* | " analyze that " is one of those crass , contrived sequels that not only fails on its own , but makes you second-guess your affection for the original . \n It was terrible \\an uneven look into a grim future that doesn't come close to the level of intelligence and visual splendour that can be seen in other films based on philip k . dick stories .\n it was terrible ...about the only thing to give the movie points for is bravado – to take an entirely stale concept and push it through the audience's meat grinder one more time .\n It was |
| SST-5 | Sentiment of the sentence: | {great, good, okay, bad, terrible } | It was *Label* | a string, funny and finally transportin re-imagining of beauty and the beast and 1930s horror films \n It was great \\apparently ressembled from the cutting-room floor of any given daytime soap. \n It was bad ...no movement, no yuks, no much of anythin. \n It was |
| Subj | Subjectivity of the sentence: | {subjective, objective } | It's *Label* | gangs , despite the gravity of its subject matter , is often as fun to watch as a good spaghetti western . \n It's subjective \\in other words , it's just another sports drama/character study . yet this one makes up for in heart what it lacks in outright newness . plus , like i already mentioned . . . it's robert duvall ! c'mon !\n it's subjective ...smart and alert , thirteen conversations about one thing is a small gem . \n It's |
| WebQ | Answer the following question. Question: <Question> \t Answer: <Answer> | N/A | N/A | Answer the following question. Question: who was dan cody?\t Answer: American football player\n Answer the following question. Question: who is james dean? \t Answer: Actor \n ...Answer the following question. Question: who created microsoft windows?\t Answer: |

Table 7 (Cont.).: Prompt and instruction used for each dataset. We denote examples as blue color, and query as red color, respectively. (Continued)

## B BENCHMARK OVERVIEW

In this paper, we employed nine text classification tasks: four for sentiment classification, two for topic classification, two for natural language inference, and one for subjectivity classification. The statistics for each dataset are provided below Table 8. To simulate the limited candidate dataset setting, we randomly sampled 200 instances from the training set as the candidate samples by default.

| Dataset | Type | Training | Test | Num class |
|---------|------|----------|------|-----------|
| SST-2 | Sentiment | 6,911 | 1,821 | 2 |
| SST-5 | Sentiment | 8.534 | 2,210 | 5 |
| AGNews | Topic | 29,914 | 3,000 | 4 |
| Subj | Subjectivity | 8,000 | 2,000 | 2 |
| MR | Sentiment | 8,662 | 2,000 | 2 |
| CR | Sentiment | 1,772 | 1,996 | 2 |
| TREC | Topic | 5,381 | 500 | 6 |
| RTE | NLI | 2,490 | 3,000 | 2 |
| MNLI | NLI | 263,689 | 9,796 | 3 |
| WebQ | QA | 3,022 | 756 | N/A |

Table 8: The statistics of the datasets used in this paper.

## C    GENERATION TASK NOISE ANNOTATION

Following Gao et al. (2024), we generate the noised annotation for WebQ by replacing the original output with random output from a large corpus with probability $p$. The large corpus is constructed by training sample outputs of NQ (Kwiatkowski et al., 2019), TREX (Elsahar et al., 2018), ProtoQA (Boratko et al., 2020), SQuAD (Rajpurkar, 2016), AY2 (Hoffart et al., 2011), SciQ (Pedersen et al., 2020), QUAREL (Tafjord et al., 2019), CommonsenseQA (Talmor et al., 2019), ARC (Clark et al., 2018) and CODAH (Alisa & Downey, 2019). We filter out outputs with length longer than 10 to make sure the candidate noised annotation follows the length distribution in the original WebQ.

## D    CASE STUDY

We provide case studies on SST-2 and SST-5 by comparing different selected ICL samples using PromptPG and PromptPG+**ConDS** in Table  9 and Table  10. As shown in these two tables, by applying **ConDS**, we increase the clean ratio in the selected ICL samples from $95\%$ to $100\%$ and from $35\%$ to $95\%$ for SST-2 and SST-5, respectively. For SST-2, although the selected ICL samples of PromptPG only include one noisy sample, the final prediction of LLM has been misled. By applying **ConDS** with simple duplication, we augment the most informative samples and filter out the misleading ones. As we can observe from the tables, samples with similar answers as the query question are more likely to be included in the selected sample set after distribution shift of the candidate pool. These most informative samples correctly guide the final prediction of the LLM to the right answer. Besides, if we conduct one more step to filter out the duplicated samples, we can even cut the query token size with a correct prediction.

| Query question | this is art paying homage to art. | |
|---|---|---|
| Retriever | PromptPG | PromptPG+**ConDS** |
| Retrieved samples | seriously , rent the disney version. Label: Terrible. | i could just feel the screenwriter at every moment ' tap , tap , tap , tap , tapping away ' on this screenplay. Label: Great. |
| | the rich performances by friel – and especially williams , an american actress who becomes fully english – round out the square edges. Label: Great. | i could just feel the screenwriter at every moment ' tap , tap , tap , tap , tapping away ' on this screenplay. Label: Great. |
| | did no one on the set have a sense of humor , or did they not have the nerve to speak up? Label: Terrible. | i could just feel the screenwriter at every moment ' tap , tap , tap , tap , tapping away ' on this screenplay. Label: Great. |
| | see it. Label: Great. | i could just feel the screenwriter at every moment ' tap , tap , tap , tap , tapping away ' on this screenplay. Label: Great. |
| | a quiet , disquieting triumph. Label: Great. | the film does a solid job of slowly , steadily building up to the climactic burst of violence. Label: Great. |
| | a raunchy and frequently hilarious follow-up to the gifted korean american stand-up 's i 'm the one that i want. Label: Great. | the film does a solid job of slowly , steadily building up to the climactic burst of violence. Label: Great. |
| | nearly every attempt at humor here is doa. Label: Terrible. | the film does a solid job of slowly , steadily building up to the climactic burst of violence. Label: Great. |
| | a lot like the imaginary sport it projects onto the screen – loud , violent and mindless. Label: Terrible. | the film does a solid job of slowly , steadily building up to the climactic burst of violence. Label: Great. |
| | elaborate special effects take centre screen , so that the human story is pushed to one side. Label: Great. | the film does a solid job of slowly , steadily building up to the climactic burst of violence. Label: Great. |
| | reyes ' directorial debut has good things to offer , but ultimately it 's undone by a sloppy script. Label: Terrible. | the film does a solid job of slowly , steadily building up to the climactic burst of violence. Label: Great. |
| | earnest yet curiously tepid and choppy recycling in which predictability is the only winner. Label: Terrible. | the film does a solid job of slowly , steadily building up to the climactic burst of violence. Label: Great. |
| | the sum of all fears is remarkably fuddled about motives and context , which drains it of the dramatic substance that would shake us in our boots (or cinema seats). Label: Terrible. | the film does a solid job of slowly , steadily building up to the climactic burst of violence. Label: Great. |
| | that rare movie that works on any number of levels – as a film of magic and whimsy for children , a heartfelt romance for teenagers and a compelling argument about death , both pro and con , for adults. Label: Great. | the film does a solid job of slowly , steadily building up to the climactic burst of violence. Label: Great. |
| | this is the kind of movie that gets a quick release before real contenders arrive in september. Label: Terrible. | the film does a solid job of slowly , steadily building up to the climactic burst of violence. Label: Great. |
| | the movie is a negligible work of manipulation , an exploitation piece doing its usual worst to guilt-trip parents. Label: Terrible. | sensitive , moving , brilliantly constructed work. Label: Great. |
| | the holes in this film remain agape – holes punched through by an inconsistent , meandering , and sometimes dry plot. Label: Terrible. | sensitive , moving , brilliantly constructed work. Label: Great. |
| | we want the funk - and this movie 's got it. Label: Great. | a moving and important film. Label: Great. |
| | blessed with a searing lead performance by ryan gosling (murder by numbers) , the movie is powerful and provocative. Label: Terrible. | a moving and important film. Label: Great. |
| | not only are the film 's sopranos gags incredibly dated and unfunny , they also demonstrate how desperate the makers of this ' we 're - doing-it-for - the-cash ' sequel were. Label: Terrible. | a moving and important film. Label: Great. |
| | the way home is an ode to unconditional love and compassion garnered from years of seeing it all , a condition only the old are privy to , and ... often misconstrued as weakness. Label: Great. | a moving and important film. Label: Great. |
| Prediction | Terrible (✗) | Great (✓) |

Table 9: Case study on SST-2 for retrieved samples of PromptPG and PromptPG+**ConDS**. The noisy samples are marked in red.

| Query question | as hugh grant says repeatedly throughout the movie. | |
|---|---|---|
| Retriever | PromptPG | PromptPG+**ConDS** |
| Retrieved samples | meant for star wars fans . Label: Okay. | While it can be found in various regions, the most striking is its remarkable level of closeness. Label: Great. |
| | very well made , but does n't generate a lot of tension. Label: Good. | A superb movie is overshadowed by sentimental cliches. Label: Good. |
| | the performers are so spot on , it is hard to conceive anyone else in their roles. Label: Great. | Whenever possible, Bill Plympton, the master animator, is available to make new films. Label: Great. |
| | the sentimental cliches mar an otherwise excellent film. Label: Okay | Whenever possible, Bill Plympton, the master animator, is available to make new films. Label: Great. |
| | not as good as the original , but what is ... Label: Good. | Whenever possible, Bill Plympton, the master animator, is available to make new films. Label: Great. |
| | however sincere it may be , the rising place never quite justifies its own existence. Label: Terrible. | Despite the sentimental messages it conveys, this fantastic movie suffers. Label: Good. |
| | the tasteful little revision works wonders , enhancing the cultural and economic subtext , bringing richer meaning to the story 's morals. Label: Good. | Despite its widespread presence, the most remarkable feature is its exceptional level of intimacy. Label: Great. |
| | not quite as miraculous as its dreamworks makers would have you believe , but it more than adequately fills the eyes and stirs the emotions. Label: Great. | In its poetic form, cantet demonstrates the delicate contrast between being inside and outside, one looking in. Label: Good. |
| | frequent flurries of creative belly laughs and genuinely enthusiastic performances ... keep the movie slaloming through its hackneyed elements with enjoyable ease. Label: Terrible. | In its poetic form, cantet demonstrates the delicate contrast between being inside and outside, one looking in. Label: Good. |
| | humor in i spy is so anemic. Label: Great. | Despite the sentimental messages it conveys, this fantastic movie suffers. However, Label: Good. |
| | it 's a good film , but it falls short of its aspiration to be a true ' epic '. Label: Bad | Despite the sentimental messages it conveys, this fantastic movie suffers. However, Label: Good. |
| | pretend like your sat scores are below 120 and you might not notice the flaws. Label: Okay. | An emotional film that resembles a documentary as it depicts the transformation of an Italian immigrant family. Label: Great. |
| | the warnings to resist temptation in this film ... are blunt and challenging and offer no easy rewards for staying clean. Label: Okay. | Even after 20 years, it retains the title of "the first genuine masterpiece" Spielberg awarded and deserved all the hearts it won. Label: Great. |
| | with " ichi the killer " , takashi miike , japan 's wildest filmmaker gives us a crime fighter carrying more emotional baggage than batman ... Label: Good. | The movie is outstanding, incorporating humor, sexuality, and sentimentality. Label: Good. |
| | call this the full monty on ice , the underdog sports team formula redux. Label: Bad. | Despite its age, Spielberg's first true masterpiece remains in high demand among many. Label: Great. |
| | a new film from bill plympton , the animation master , is always welcome. Label: Bad. | An excellent film is marred by sentimental clichés. Label: Good. |
| | the emperor 's club is one of those films that possesses all the good intentions in the world , but ... Label: Good. | A cleverly crafted film that is both entertaining and skillfully executed. Label: Great. |
| | a lyrical metaphor for cultural and personal self-discovery and a picaresque view of a little-remembered world. Label: Bad. | The movie is skillfully executed and enjoyable, with skilled acting and direction. Label: Great. |
| | may prove to be ( tsai 's ) masterpiece. Label: Great. | The transformation of an Italian immigrant family is the subject of a moving film that has hints at documentary quality. Label: Great. |
| | a great cast and a wonderful but sometimes confusing flashback movie about growing up in a dysfunctional family. Label: Good. | The film is a work of art and entertainment, with well-crafted acting and direction. Label: Great. |
| Prediction | Bad (✗) | Great (✓) |

Table 10: Case study of SST-5 for retrieved samples of PromptPG and PromptPG+**ConDS**. The noisy samples are marked in red.