# OpenReview forum: "ConDS: Context Distribution Shift for Robust In-Context Learning"
_ICLR.cc/2025/Conference — Submitted to ICLR 2025_

### Official Review · Reviewer_21KR · 2024-10-24

**Soundness:** 2
**Presentation:** 2
**Contribution:** 2
**Rating:** 5
**Confidence:** 4

**Summary:**

The paper introduces ConDS to handle noisy ICL examples which could be misleading and result in degraded ICL performance. ConDS tackles this by adjusting the distribution of the candidate pool —identifying clean, informative examples through retriever scores and LLM feedback, then boosting them while downplaying noisy ones. The paper also mathematically proved that this process is equivalent to dynamically fine-tuning a retriever. Rather than developing a new retriever, ConDS enhances the data for existing retrievers like BM25, KNN, and fine-tuned ones like PromptPG. The paper’s experiments show ConDS improves performance significantly—by about 8.12%—across various tasks like sentiment analysis and topic classification, particularly in noisy conditions. The key takeaway is ConDS boosts ICL’s reliability by ensuring cleaner samples are used during learning.

**Strengths:**

The paper introduces a practical approach to improve ICL run-time robustness by adaptively adjusting the distribution of the demonstration pool. This approach is not limited to the choice of example retriever, i.e. off-the-shelf and fine-tuned retrievers can both be integrated, making the system flexible. It also shows overall promising results on several benchmarks (8.1% average performance boost), and especially in noisy data environments.

**Weaknesses:**

Although the paper recognize a real-world problem - contamination in ICL example pool and developed a practical mitigation strategy, it is still somewhat incremental and I am questionable about it's generalizability. There are lots of other real-world complexities that have not been considered.
1. The datasets assessed here are not very challenging, mostly classification. It's uncertain how this behaves on more challenging use-cases such as text2sql, RAG, plus the binary signal used to distinguish between noisy / informative examples can be hard to generalize on other tasks.
2.  The inference model used here is a fairly out-dated model GPT-neo-2.7B., and whether such method will still be effective towards a more powerful llm is unclear.
3. The "training" stage is not very scalable as the pool size increases and the queries are long.
4. The definition of noise: looking at the noise example provided, the labels are completely irrelevant with the ground truths. In real world scenario, noise can be more nuanced and there lacks of discussion how to handle borderline cases (e.g. when examples are ambiguous)

**Questions:**

1. Have you tested the transferability across other tasks?
2. How would this approach over-penalize borderline examples that may actually hold some useful contextual information? In complex tasks such as function calling, code generation, maybe the label contents do not match exactly with ground truth, but the formatting can be useful? Also what would happen when the query is challenging and hard to achieve good performance by adding :relevant good examples and those good examples are marked as "problematic" ones?
3. The paper mentions using simple duplication or paraphrasing for augmenting clean examples. Although it might not be the focus of this paper - have you considered other augmentation methods such as adversarial example generation, i.e. adding noise to $x_i^{k}$ in the retrieved example (not $y_i^k$), to not only reduce noise examples but enhance the quality of the informative examples?
4. Regarding scalability, any profiling of training time regarding dataset size and LLM size?

---

> ### Author Response · Authors · 2024-11-24
> **Response for your comments and suggestions - Part 1**
>
> **1. The datasets assessed here are not very challenging, mostly classification.**
>
> The effects of noisy samples for in-context learning (ICL) remain underexplored by prior arts. Thus, in this paper, we take a first step to propose the context distribution shift method to tackle noisy types including mislabelling for classification tasks and wrong answers for question answering (QA) tasks. We also supplement more QA task results in section 4.5. The possible noise type for other tasks or their possible solutions remains unexplored, but we think it would be very interesting and worthwhile directions to continue exploring in the future!
>
> **2. The inference model used here is a fairly out-dated model GPT-neo-2.7B.**
>
> Thank you very much for your suggestion! We supplemented section 4.5 with more results using Llama2-7B as our inference model. According to the results, we can summarize the following findings. First, irrelevant noisy information can cause performance degradation for baselines, which becomes more severe as the noise ratio increases (e.g., $0.1521 \to 0.0529$ in BM25 case for WebQ). Second, as shown in bold font, ConDS demonstrates improved performance compared to other retrieval methods. For the case when the noise ratio is 0.4, our method outperformed the best baseline by $8\%$ and $11.9\%$, respectively for two datasets. This indicates that focusing on clean samples can mitigate the performance decline caused by noise ICL samples. Moreover, even with an increased noise ratio $0.2 \to 0.4$, \alg shows a stable performance $0.1600 \to 0.1650$ and $0.3590\to0.3870$ without a degradation. Consequently, utilizing ConDS is a robust method for both classification and generation tasks when ICL dataset has noise information.
>
> **3. The "training" stage is not very scalable as the pool size increases and the queries are long.**
>
> We want to clarify that the pool size will not continually increase. As shown in Algorithm 1 line15-16, we will randomly subsample $N_{upp}$ samples from the candidate pool, if the pool size exceeds $N_{upp}$.
>
> The length of the queries does not get longer with the training process either, since we fix the number of shots for each query. We set the shot number as $20$ by default.
>
> **4. The definition of noise**
>
> We agree with you that there exists different kinds of noises, but among them label noise is an important problem worth exploring both in NLP [A-C] and and other domains. To show the generalization of our method, we also explore a similar case as label noise, which is wrong answer for question answering (QA) tasks in section 4.5.
>
> [A] Jannik Kossen, Yarin Gal, and Tom Rainforth. In-context learning learns label relationships but is not conventional learning. ICLR 2024.
>
> [B] Jerry Wei, Jason Wei, Yi Tay, Dustin Tran, Albert Webson, Yifeng Lu, Xinyun Chen, Hanxiao Liu, Da Huang, Denny Zhou, et al. Larger language models do in-context learning differently. arXiv preprint arXiv:2303.03846, 2023.
>
> [C] Chen Cheng, Xinzhi Yu, Haodong Wen, Jinsong Sun, Guanzhang Yue, Yihao Zhang, and Zeming Wei. Exploring the robustness of in-context learning with noisy labels. arXiv preprint arXiv:2404.18191,2024.
>
> **5. Have you tested the transferability across other tasks?**
>
> We add more results in section 4.5 for the generation tasks: question answer (QA).
>
> **6. How would this approach over-penalize borderline examples that may actually hold some useful contextual information? In complex tasks such as function calling, code generation, maybe the label contents do not match exactly with ground truth, but the formatting can be useful? Also what would happen when the query is challenging and hard to achieve good performance by adding :relevant good examples and those good examples are marked as "problematic" ones?**
>
> Thanks for your comments! The purpose of method is to selecting the most informative samples by using the feedback from the LLM as the guidance to changing the distribution of context samples. Thus, if the feedback from the LLM is positive, that means the retrieved samples can be useful, we do not care if the positive effect of this sample is due to the correct label or the format as you mentioned in your comments or other aspects. As long as these samples show positive effects, we will augment this part of samples. According to our experimental results, these informative samples are more likely to be clean samples as shown in the case study in Table 10 and Table 11 in the appendix. We also found that samples with similar answers as the query question are more likely to be informative samples. The informative samples may vary according to different tasks.
>
> We focus on classification task and QA in this paper. For other tasks, such as function calling, they do not have labels, and the noise type for these tasks remains unexplored by prior arts. Thus, we leave the possible noise type for other tasks or their possible solutions for future work.

---

> ### Author Response · Authors · 2024-11-24
> **Response for your comments and suggestions - Part 2**
>
> **7. The paper mentions using simple duplication or paraphrasing for augmenting clean examples. Although it might not be the focus of this paper - have you considered other augmentation methods such as adversarial example generation, i.e. adding noise to $x_i^k$ in the retrieved example (not $y_i^k$), to not only reduce noise examples but enhance the quality of the informative examples?**
>
> Thanks for your comment! We suppose the method you mentioned here is one kind of denoising method for the question ($x_i^k$), but denoising method will not work for the noise type we investigated in our paper including label noise for classification tasks or wrong answers for QA. Denoising methods for nlp [A-C] are targeted at linguistic noise such as grammar correction rather than other types of noise (such as mislabelling or wrong answers). The goal of data denoising is to remove unwanted information from the text. Removing this part of noise will not work for our problem setting.
>
> [A] Al Sharou K, Li Z, Specia L. Towards a better understanding of noise in natural language processing[C]//Proceedings of the International Conference on Recent Advances in Natural Language Processing (RANLP 2021). 2021: 53-62.
>
> [B] Freitag M, Roy S. Unsupervised natural language generation with denoising autoencoders[J]. arXiv preprint arXiv:1804.07899, 2018.
>
> [C] Xie Z, Genthial G, Xie S, et al. Noising and denoising natural language: Diverse backtranslation for grammar correction[C]//Proceedings of the 2018 Conference of the North American Chapter of the Association for Computational Linguistics: Human Language Technologies, Volume 1 (Long Papers). 2018: 619-628.
>
> **8. Regarding scalability, any profiling of training time regarding dataset size and LLM size?**
>
> The training time will not be affected by the dataset size, since the candidate pool size will not continually increase, and will remains under the upper bound we preddfined (see Algorithm 1 line 15-16). The validation dataset size is also fixed for each epoch.
>
> The training time is mostly decided by the query time of the LLM itself. The larger the model, the longer the query time, but our method will introduce almost no extra time cost comparing with existing fine-tuned retrievers. As shown in line 461-462, for SST-2 using GPT-Neo-2.7B, the training time for $1$ epoch of PromptPG and PromptPG+ConDS is $12$ and $13$ seconds, respectively. The extra time is negligible.

---

> > ### Comment · Reviewer_21KR · 2024-11-24
> >
> > I thank the authors for their response that addressed some of my questions. I still think to support the claim "ConDS improve the ICL performance" needs more task coverage rather than simple classification and QA. Therefore, I increased my rating from 3 -> 5. Good luck.

---

> > > ### Author Response · Authors · 2024-11-24
> > >
> > > Thank you very much for carefully reading our response and raising your score! We are glad our response has addressed some of your concerns.

---

### Official Review · Reviewer_mTiq · 2024-10-31

**Soundness:** 3
**Presentation:** 3
**Contribution:** 3
**Rating:** 6
**Confidence:** 4

**Summary:**

This paper introduces ConDS, an approach designed to filter noisy in-context examples from a candidate set using LLM feedback—in this case, the prediction of the LLM on a held-out split of the candidate set—to distinguish between noisy and non-noisy examples. The method is straightforward and effective, demonstrating notable improvements over the strongest baseline, PromptPG, evaluated in this study.

However, it is worth noting that the paper does not address why existing LLM feedback-based filtering methods, which employ similar entropy/perplexity based feedback mechanisms, cannot be directly applied in noisy settings.

**Strengths:**

1. Demonstrates a significant performance improvement over baselines in noisy settings, showing proposed approach's effectiveness in filtering noisy in-context examples.
2. An additional strength lies in the static approach's simplicity, as it can be seamlessly applied to any in-context pipeline with minimal modifications.

**Weaknesses:**

1. There is lack of clarity in contextualizing this work against prior studies on filtering in-context demonstrations. Although these existing methods operate in non-noisy settings, many rely on LLM feedback [1,2,3], often in the form of entropy or perplexity, similar to ConDS. Clarifying why such methods are not discussed would be beneficial. [3] originally is applied to find the best order of the prompt but it can potentially be used to provide the weighting of each in-context example in the noisy setting.

2. UDR [4], mentioned in related work, also fine-tunes a retriever based on LLM feedback, yet it is unclear why training a UDR-style model on feedback is not included.

**Questions:**

**Questions and Comments**

1. Consider ConE: ConE [3] appears to be applicable for re-weighting the candidate set $C^{\text{train}}$ based on the informativeness of retrieved examples, as different prompts of in-context examples would have higher perplexity. ConDS and ConE share similarities in this respect, but there is no discussion on these parallels.
2. Comparing PromptPG + ConDS with UDR: Based on Lemma 1, how does the combination of PromptPG + ConDS differ from training a UDR-style model on the target task? Since PromptPG + ConDS also requires retriever training on the target task, it would seem that a UDR-like method, which incorporates LLM feedback directly into the retriever's fine-tuning, would serve as a useful baseline.
3. Noise Ratio in Figure 5(a): Why is the maximum noise ratio capped at 0.6? It would be insightful to know if ConDS can filter noise effectively at even higher noise levels, which may align with noisy samples in the validation split of $C^{\text{train}}$.
4. Definition of SCORE($\cdot$): It is not specified what SCORE($\cdot$) represents. Are these similarity scores from the retriever?
5. Static Augmentation with ConDS: Are there results on applying ConDS to PromptPG in the static augmentation scenario? I am assuming that all other values in Table 2 are from the static setting.
6. The term ‘augmentation time’ is confusing, as it actually refers to the number of augmentations after upsampling. Consider renaming it to ‘augmentation size’ or an equivalent term for clarity.

**Typos**
- Line 131: "concatination" → "concatenation"
- Line 533: "as followings" → "as follows"
- Algorithm 1, Lines L6 and L7: Should $q_i$ be $x_i$?

**References**

[1] Demystifying Prompts in Language Models via Perplexity Estimation (Gonen et al., EMNLP Findings 2023)

[2] Revisiting Demonstration Selection Strategies in In-Context Learning (Peng et al., ACL 2024)

[3] Fantastically Ordered Prompts and Where to Find Them: Overcoming Few-Shot Prompt Order Sensitivity (Lu et al., ACL 2022)

[4] Unified Demonstration Retriever for In-Context Learning (Li et al., ACL 2023)

---

> ### Author Response · Authors · 2024-11-24
> **Response for your comments and suggestions**
>
> We are glad that the reviewer found our proposed approach effective and practical. We thank the reviewer for the constructive comments and suggestions, which we address below:
>
> **1. Lack of UDR-style methods based on LLM feedback**
>
> Our ConDS method is a context distribution shift method, which can be used to enhance the robustness of different retrievers including off-the-shlf ones and fine-tuned ones against noisy samples. Since UDR and PromptPG are both fine-tuned retriever methods with similar styles based on the feedback from the LLM as you mentioned, we use PromptPG as a representative method of this kind of retrievers to show the enhancement of our ConDS for this kind of retrievers. We believe ConDS can also be combined with other UDR-style retrievers [1,2,3,4] using a similar strategy we adopt for PromptPG+ConDS, and we will explore ConDS for similar styles retrievers in future works.
>
> **2. Noise Ratio in Figure 5(a): Why is the maximum noise ratio capped at 0.6? It would be insightful to know if ConDS can filter noise effectively at even higher noise levels, which may align with noisy samples in the validation split of Ctrain.**
>
> We add results for SST-2 when noise ratio = 0.8 in the following table. With noise ratio=0.8, ConDS still outperformed the baselines.
>
> |Method|KNN|BM25|PromptPG|PromptPG+ConDS|
> | --- | --- |--- |--- |--- |
> |Acc|0.5272|0.5360|0.8320|**0.8726**|
>
> **3. Definition of SCORE(⋅): It is not specified what SCORE(⋅) represents. Are these similarity scores from the retriever?**
>
> Yes, SCORE() represents different ranking scores produced by different retrievers for the candidate samples, most retriever adopt different similarity scores. We will give clearer definition in our revised manuscript.
>
> **4. Static Augmentation with ConDS: Are there results on applying ConDS to PromptPG in the static augmentation scenario? I am assuming that all other values in Table 2 are from the static setting.**
>
> For off-the-shelf retrievers including BM25, KNN, DPP in Table 2, the augmentation is static. For the fine-tuned retrievers such as PromptPG in Table 2 and Table 1, since the retriever is trained and updated, the augmentation is dynamic.
>
> **5. The term ‘augmentation time’ is confusing, as it actually refers to the number of augmentations after upsampling. Consider renaming it to ‘augmentation size’ or an equivalent term for clarity.**
>
> Thanks for your suggestion! To avoid misunderstanding, we rename ‘augmentation time’  as ‘augmentation size’ in our revised manuscript.
>
> **6. We have corrected all the typos in our revised manuscript.**

---

> ### Comment · Reviewer_mTiq · 2024-11-24
>
> I thank the authors for their response, which clarifies some of my concerns. The following questions still remain:
>
> 1. How does ConDS contrast against existing LLM-feedback based filtering methods, such as ConE?
>
> > We believe ConDS can also be combined with other UDR-style retrievers
>
> 2. I understand and agree with the authors that ConDS can be applied to any retriever. I would like to clarify my original question. Why cannot the training procedure of UDR be directly be applied to fine-tune a retriever (without ConDS), since it uses LLM feedback by default? I would assume this would be a suitable baseline (question 2 in my review).

---

### Official Review · Reviewer_TpoZ · 2024-11-03

**Soundness:** 2
**Presentation:** 3
**Contribution:** 2
**Rating:** 3
**Confidence:** 4

**Summary:**

The paper studies in-context learning (ICL) where the pool of examples includes noisy examples. To address this challenge, the paper proposes ConDS, which focuses on improving ICL robustness. ConDS identifies clean and informative samples based on the validation set, and then removes noisy examples that contribute to negative performance. Experimental results on nine datasets show ConDS's robustness on noisy ICL examples.

**Strengths:**

The strengths of the paper are outlined below:

- S1)  The paper examines the robustness of ICL, offering new insight for various LLM-based applications.
- S2) ConDS significantly outperforms competing baselines in noisy conditions.
- S3) The motivation is clear, and the paper is easy to follow.

**Weaknesses:**

The weaknesses of the paper are outlined below:

-  W1) I have some concerns regarding the methodology. ConDS relies on the validation set to classify examples as clean or noisy. However, since the validation set itself may contain noise, this could lead to inaccurate predictions. How do the authors ensure that the feedback from the validation set is reliable?

- W2) The experimental setup and results are unconvincing. The default noise ratio is set to $p=0.6$, which results in the majority of the pool being noisy. In this scenario, it would be reasonable to conduct zero-shot inference using advanced LLMs, such as LLaMA-3, and disregard the noisy pool entirely. However, the authors only test smaller, outdated models like GPT-Neo-2.7B, which do not provide meaningful insights into zero-shot performance. Could the authors present zero-shot results for more advanced models of various sizes?

- W2) ConDS seems to be an extension of PromptPG, which may limit its broader applicability (although ConDS can be combined with other retrievers, its performance is suboptimal). Could the authors elaborate on the unique contributions of this work?

**Questions:**

Some additional questions/comments are outlined below:

- Q1) Could you further clarify the differences between Sections 3.1 and 3.2? In Section 3.1, you utilize a paraphrasing model, while in Section 3.2, you employ a fine-tuned retriever to define $E_{shift}$. Is this correct and how do the two approaches compare in terms of performance?

- Q2) The paraphrasing model is a T5 model trained on ChatGPT responses. Could you augment the baselines with this model and achieve better performance?

---

> ### Author Response · Authors · 2024-11-24
> **Response for the weakness**
>
> We are glad that the reviewer found our method new and our experimental results significant. We thank the reviewer for the constructive comments and suggestions, which we address below:
>
> **W1) concerns regarding the methodology: How do the authors ensure that the feedback from the validation set is reliable?**
>
> Due to the existence of noisy samples in the validation set, after data augmentation and subsampling, a small percentage of noisy sample still exists in the candidate pool as shown in Figure 3b. Our method is not to completely filter out all noisy samples, but to increase the clean sample ratio and change the sample distributions. We explain in more detail as follows:
>
> The original candidate set distribution is shown in Figure 3a. The clean and noisy samples are mixed in the candidate set. During ICL, the retriever tends to select similar samples to the query as the ICL samples. With mixed clean and noisy samples, sampling similar samples using the retriever easily includes both clean and noisy samples for almost all query samples.
>
> The distribution of the candidate pool after ConDS is shown in Figure 3b. Instead of mixing clean and noisy samples, the neighbors of the clean samples are also augmented with more clean samples. During the inference stage, the retriever tends to select the most relevant samples for the test queries. The most relevant spaces are filled with clean samples, and the misleading samples tend to have a lower relevance score. Misleading sample embeddings stay far away from the clean samples cluster, so they will not interfere with the test queries lying close to the clean samples. Hence, we reduce the catastrophic impact of the noisy samples from almost all test queries to only a small percentage of queries. This visualization results intuitively explain why our method works. This part of the explanation can also be found in line 204-213 in the paper.
>
> **W2) The experimental setup and results are unconvincing. Could the authors present results for more advanced models?**
>
> Thank you very much for your suggestion. We agree with you that classification tasks would be two simple for larger LLMs using 0-shot, thus, we use a use larger LLM Llama2-7B as our inference model, and test on more complicated question answering (QA) tasks: WebQ [A] and Squad [B] in section 4.5. Note that, Llama2-7B using 0-shot inference can only achieve very low accuracy on these two QA tasks.
>
>  According to the results, we can summarize the following findings. First, irrelevant noisy information can cause performance degradation for baselines, which becomes more severe as the noise ratio increases (e.g., $0.1521 \to 0.0529$ in BM25 case for WebQ). Second, as shown in bold font, ConDS demonstrates improved performance compared to other retrieval methods. For the case when the noise ratio is 0.4, our method outperformed the best baseline by 8\% and 11.9\%, respectively for two datasets. This indicates that focusing on clean samples can mitigate the performance decline caused by noise ICL samples. Moreover, even with an increased noise ratio $0.2 \to 0.4$, \alg shows a stable performance $0.1600 \to 0.1650$ and $0.3590\to0.3870$ without a degradation. Consequently, utilizing ConDS is a robust method for both classification and generation tasks when ICL dataset has noise information.
>
> [A] Jonathan Berant, Andrew Chou, Roy Frostig, and Percy Liang. Semantic parsing on freebase from question-answer pairs. In Proceedings of the 2013 conference on empirical methods in natural language processing, pp. 1533–1544, 2013.
>
> [B] P Rajpurkar. Squad: 100,000+ questions for machine comprehension of text. arXiv preprint arXiv:1606.05250, 2016.
>
> **W3) Could the authors elaborate on the unique contributions of this work?**
>
> We summarize our contributions as follows:
>
> 1.  We propose ConDS, which improves the quality of the candidate set by not only emphasizing informative samples but also reducing the impact of noisy label samples. We are the first to investigate the power of distribution shift of the candidate set to improve the ICL performance.
>
> 2. ConDS supports different kinds of off-the-shelf and fine-tuned retrievers to enhance their robustness against noisy samples. We also provide an analysis to reveal the essential commonality between ConDS and the existing retrievers.
> As shown in Table 2, for different retrievers, we can observe an average improvement of 1.26\%, 3.36\%, 5.54\%, 6.83\%, and 9.77\% for five different retrievers (the improvement is not suboptimal), respectively, which shows that our ConDS can be flexibly combined with different kinds of retrievers. The more capable the retriever is, the more boosts we get for the ICL performance. For future more advanced retrievers, our proposed method can also further enhance their capability.

---

> ### Author Response · Authors · 2024-11-24
> **Response for the questions**
>
> **Q1) Could you further clarify the differences between Sections 3.1 and 3.2? In Section 3.1, you utilize a paraphrasing model, while in Section 3.2, you employ a fine-tuned retriever to define Eshift. Is this correct and how do the two approaches compare in terms of performance?**
>
> **Difference of section 3.1 and 3.2:** The existing retrievers can be catogorized into off-the-shelf (frozen) retriever, such as KNN, BM25, and fine-tuned retrievers, such as PromptPG. For off-the-shelf retriever the ranking score of the retriever is static, so the augmentation process based on the ranking score is also static for each epoch. For this case, only the distribution of candidate pool $E$ is dynamic. For fine-tuned retriever, since the retriever model is updated, the ranking score obtained from the updated retriever is also dynamic, so the augmentation process is also dynamic for each epoch. For this case, both the candidate pool and the augmentation process is dynamic. Section 3.1 can be considered as a basic case of Section 3.2.
>
> **Clarification of the paraphrasing model and the retriever:** The purpose of the paraphrasing model and the retriever is different, so they are not comparable. In-context-learning (ICL) operates by presenting LLMs with a set of selected ICL examples relevant to the test query from the candidate dataset $C$, preconditioning the models for the target task. The retriever is used to retrieve relevant samples from the candidate pool for ICL. The paraphrasing model is not a retriever, it is a model used for augmentation. As we mentioned in section 3.1, we can either choose directly duplicate or paraphrasing as the augmentation method once we have decided which samples should be augmented and what is the augmentation size.
>
> **Q2) The paraphrasing model is a T5 model trained on ChatGPT responses. Could you augment the baselines with this model and achieve better performance?**
>
> Thank you very much for your suggestion! We add both ConDS (duplicate) and the ConDS (paraphrase) results in Table 1 in the revised manuscript. According to the results, ConDS (duplicate) outperformed zero-shot learning by 17.07\%, and the best baseline by 8.12\% on average. ConDS (parahprase) outperformed zero-shot learning by 15.20\%, and the best baseline by 6.25\% on average. For most datasets, both augmentation method achieves either the best or second-best performance. These results indicate that the distribution shift induced by ConDS can improve the robustness of ICL no matter what augmentation method is adopted.

---

> > ### Comment · Reviewer_TpoZ · 2024-11-27
> >
> > Thank you for your efforts in responding to my comments. After careful consideration, I stand by my initial evaluation. I believe the experimental setup—particularly the pool of ICL examples—introduces a significant amount of noise, which could hinder drawing reliable conclusions. This issue also affects the validation set used by the ConDS methodology. Given this, it may be more effective to explore zero-shot inference or refine the prompt design for the LLM to improve performance.
> >
> > I recommend that the authors provide a more comprehensive evaluation, comparing the performance of ConDS with zero-shot inference across both classification and generation tasks (the new experiments are promising). Expanding the analysis in these areas would help clarify the applicability of ConDS.

---

### Official Review · Reviewer_vsUD · 2024-11-04

**Soundness:** 2
**Presentation:** 3
**Contribution:** 2
**Rating:** 5
**Confidence:** 3

**Summary:**

This paper proposes **ConDS (Context Distribution Shift)** to enhance the robustness of **In-Context Learning (ICL)** when dealing with noisy samples. The core idea is to modify the distribution of the candidate sample set to amplify informative samples and reduce the impact of misleading samples. The ConDS method primarily consists of the following steps: 1. Identifying Informative Samples: Using feedback from large language models (LLMs) and ranking scores from retrievers to identify information-rich samples within the candidate set. 2. Enhancing Informative Samples: Amplifying Informative samples by duplicating or paraphrasing them, thereby increasing their presence in the candidate set. 3. Subsampling: Conducting subsampling on the enhanced candidate set to control its size and further increase the probability of selecting Informative samples. The paper validates the effectiveness of the ConDS method through experiments on various text classification tasks, with results indicating that ConDS improves ICL performance in the presence of noisy samples. Additionally, the paper analyzes the effectiveness of combining ConDS with different retrievers, and finds that ConDS can be effectively combined with various retrievers.

**Strengths:**

1. The paper addresses the impact of noisy samples in In-Context Learning (ICL), which is a practical and important issue.
2. The experiment results seem promising. Experimental results show that the ConDS method achieves performance recovery across various text classification tasks, consistently outperforming pure retrieval baselines, in both off-the-shelf and fine-tuned retrievers setting.
3. The paper conducts extensive experiments to validate the effectiveness of the ConDS method, providing detailed analyses of the impact of different parameters.

**Weaknesses:**

1. **Lack of comparison with other denoising methods**: The paper would benefit from comparing ConDS with other dataset denoising methods.
2. **Insufficient explanations in some places**:
   - The definitions of "informative samples" and "misleading samples" are vague, lacking a thorough discussion regarding their relationships with clean and noisy samples.
   - The authors introduce the mixed score and assert that it enhances the retriever's ability to select clean samples. However, there is no experimental evidence provided to support this claim. It would be beneficial for the authors to design experiments comparing the impacts of different scoring mechanisms (e.g., using only retriever ranking scores, only sampling probabilities, and using mixed scores) on ICL performance to validate the effectiveness of the mixed score.
3. **Lack of discussion on mathematical assumptions**:  The conditions for applying the hypergeometric distribution in line 273 may need more discussion. ConDS utilizes enhancement and subsampling to modify the size and distribution of the candidate sample set, which does not strictly meet the conditions for sampling without replacement. Furthermore, the retriever does not make binary decisions but instead ranks and selects samples based on scores.
4. **Lack of case studies**:  The paper would benefit from the inclusion of case studies that illustrate the application and effectiveness of the ConDS method in experiment datasets.
5. **Lacks results of larger and more advanced LLMs**:  The experimental conclusions do not encompass larger or more advanced language models. Given that models with varying parameter sizes and training methodologies may yield different ICL results, it would be valuable for the authors to conduct further experiments involving these models to provide a more comprehensive evaluation.

**Questions:**

1. **Alternative indicators for sample selection**:  Besides LLM answer consistency, what other methods can guide the selection of samples? Have you validated the effectiveness of any other indicators in this context?
2. **Limitations observed in figure 4d**:  In Figure 4d, there are nearly 500 test queries where the clean sample ratio is 0 after applying ConDS. Does this indicate some limitations of the ConDS method? How do you plan to address and overcome these limitations in future work?

Other questions please see above weakness for reference.

---

> ### Author Response · Authors · 2024-11-24
> **Response for the weakness**
>
> We are glad that the reviewer found our problem setting important and our experiment results promising. We thank the reviewer for the constructive comments and suggestions, which we address below:
>
> **1. Lack of comparison with other denoising methods.**
>
> Thank you very much for your comment! The noise we investigated in our paper indicates label noise for classification tasks or wrong answers for QA. Denoising methods for nlp [A-C] are targeted at linguistic noise such as grammar correction rather than other types of noise (such as mislabelling or wrong answers). The goal of data denoising is to remove unwanted information from the text. Removing this part of noise will not work for our problem setting.
>
> [A] Al Sharou K, Li Z, Specia L. Towards a better understanding of noise in natural language processing[C], 2021.
>
> [B] Freitag M, Roy S. Unsupervised natural language generation with denoising autoencoders[J], 2018.
>
> [C] Xie Z, Genthial G, Xie S, et al. Noising and denoising natural language: Diverse backtranslation for grammar correction[C], 2018.
>
> **2. The definitions of "informative samples" and "misleading samples" are vague, lacking a thorough discussion regarding their relationships with clean and noisy samples. & Lack of case studies**
>
> To better show what kind of samples are informative samples and what kind of samples are misleading samples, we add case studies for retrieved samples of PromptPG and PromptPG+ConDS in Table 10 and Table 11 in section D of the appendix as you suggested. According to the right column of the table, the informative samples should be clean and tend to have similar answers to the query question. The informative samples can correctly guide the final prediction of the LLM to the right answer. According to the left column of the table, the misleading samples are more likely to be noisy samples (marked in red), even if they are clean samples, they tend to have completely different answers as the query question. Thus, the final prediction can be misled by these samples.
>
> **3. The authors introduce the mixed score and assert that it enhances the retriever's ability to select clean samples. However, there is no experimental evidence provided to support this claim.**
>
> To compare the effects of retriever ranking scores only and mixed scores with ConDS, we first show the best baseline PromptPG and PromptPG+ConDS in Table 1, and then show other retrievers and retrievers+ConDS in Table 2. According to Table 2, for different retrievers, we can observe an average improvement of 1.26%, 3.36%, 5.54%, 6.83%, and 9.77%, respectively, which shows that our ConDS can be flexibly combined with different kinds of retrievers. The more capable the retriever is, the more boosts we get for the ICL performance. The hybrid ranking score amplifies the effect of the original retriever on selecting clean samples.
>
> **4. Clarification of mathematical assumptions: the subsampling process.**
>
> The subsamlping process is not conducted at the same time as the augmentation process. As shown in Algorithm 1, we first conduct the augmentation process (line 5-14), and then a subsampling process is conducted (line 15-17), so there is no replacement during the sampling process.
>
> The subsampling is not based on the scores, we adopt random sampling instead. The score is used for the augmentation process (line 5-14 in Algorithm 1), and the subsampling (line 15-17) is a binary decision.
>
> **5. Lacks results of larger and more advanced LLMs.**
>
> Thank you very much for your suggestion. We supplementaled section 4.5 with more results using Llama2-7B as our inference model. According to the results, we can summarize the following findings. First, irrelevant noisy information can cause performance degradation for baselines, which becomes more severe as the noise ratio increases (e.g., $0.1521 \to 0.0529$ in BM25 case for WebQ). Second, as shown in bold font, ConDS demonstrates improved performance compared to other retrieval methods. For the case when the noise ratio is 0.4, our method outperformed the best baseline by 8% and 11.9%, respectively for two datasets. This indicates that focusing on clean samples can mitigate the performance decline caused by noise ICL samples. Moreover, even with an increased noise ratio $0.2 \to 0.4$, \alg shows a stable performance $0.1600 \to 0.1650$ and $0.3590\to0.3870$ without a degradation. Consequently, utilizing ConDS is a robust method for both classification and generation tasks when ICL dataset has noise information.

---

> ### Author Response · Authors · 2024-11-24
> **Response for the questions**
>
> **1. Alternative indicators for sample selection: Besides LLM answer consistency, what other methods can guide the selection of samples? Have you validated the effectiveness of any other indicators in this context?**
>
> Since our work is the first to investigate the power of distribution shift of the candidate set to improve the ICL performance, we do not have existing works to guide the selection of augmented samples. In our experiments, we found that using LLM answer consistency is a good way to indicate what kind of samples should be augmented. We have also tried to use the perplexity score or attention scores as the criteria, but they do not show promising results, so we leave the exploration of other indicators for future works.
>
> **2. Limitations observed in figure 4d: In Figure 4d, there are nearly 500 test queries where the clean sample ratio is 0 after applying ConDS. Does this indicate some limitations of the ConDS method? How do you plan to address and overcome these limitations in future work?**
>
> For future works, we plan to address this limitation by replacing random sampling with Clustered Sampling. As shown in Figure 3b, after the context distribution shift, both clean samples and noisy samples are clustered together. Thus, we first cluster the training set embedding $z$ into $M$ clusters. Then we try to drop out clusters that are mostly composed of noise samples (red ones). To achieve this goal, we randomly sample a few examples from each cluster to conduct $0$-shot inference for the LLM. If most of the answers given by the LLM are different from the provided answer, we drop out these clusters. We suppose the remaining cluster number is $M'$. In this way, we can furture filter out noisy samples.

---

### Meta-Review · Area_Chair_Thgf · 2024-12-13

**Metareview:**

In this paper, the authors proposed a new approach to make ICL more robust by reducing the impact of misleading samples.

There are some major concerns raised by the reviewers regarding the proposed methodology, such as the sample selection algorithms, the quality of the validation set, etc. The reviewers also raised concerns about the experimental setup and results.

The authors failed to address the original concerns as well as the follow-up questions raised by the reviewers.

Therefore, this paper is not ready for publication.

**Additional Comments On Reviewer Discussion:**

Some follow-up questions are asked by the reviewers regarding some detailed designs of the proposed method and empirical studies. The authors failed to respond.

---

### Decision · Program_Chairs · 2025-01-22

Reject